# Expression of Lineage Transcription Factors Identifies Differences in Transition States of Induced Human Oligodendrocyte Differentiation

**DOI:** 10.3390/cells11020241

**Published:** 2022-01-11

**Authors:** Florian J. Raabe, Marius Stephan, Jan Benedikt Waldeck, Verena Huber, Damianos Demetriou, Nirmal Kannaiyan, Sabrina Galinski, Laura V. Glaser, Michael C. Wehr, Michael J. Ziller, Andrea Schmitt, Peter Falkai, Moritz J. Rossner

**Affiliations:** 1Department of Psychiatry and Psychotherapy, University Hospital, LMU Munich, 80336 Munich, Germany; Florian.Raabe@med.uni-muenchen.de (F.J.R.); marius.stephan@med.uni-muenchen.de (M.S.); Benedikt.Waldeck@med.uni-muenchen.de (J.B.W.); Venena.Huber@med.uni-muenchen.de (V.H.); Damianos.Demetriou@med.uni-muenchen.de (D.D.); Nirmal.Kannaiyan@med.uni-muenchen.de (N.K.); Sabrina.Galinski@med.uni-muenchen.de (S.G.); Michael.Wehr@med.uni-muenchen.de (M.C.W.); Andrea.Schmitt@med.uni-muenchen.de (A.S.); Peter.Falkai@med.uni-muenchen.de (P.F.); 2International Max Planck Research School for Translational Psychiatry (IMPRS-TP), 80804 Munich, Germany; 3Systasy Bioscience GmbH, 81669 Munich, Germany; 4Department of Computational Molecular Biology, Max Planck Institute for Molecular Genetics, 14195 Berlin, Germany; glaser@molgen.mpg.de; 5Max Planck Institute of Psychiatry, 80804 Munich, Germany; michael_ziller@psych.mpg.de; 6Department of Psychiatry, University of Münster, 48149 Münster, Germany; 7Laboratory of Neurosciences (LIM-27), Institute of Psychiatry, University of São Paulo (USP), São Paulo 05403-903, Brazil

**Keywords:** directed differentiation, oligodendrocytes, human pluripotent stem cells, hiPSC, scRNAseq, RNA velocity

## Abstract

Oligodendrocytes (OLs) are critical for myelination and are implicated in several brain disorders. Directed differentiation of human-induced OLs (iOLs) from pluripotent stem cells can be achieved by forced expression of different combinations of the transcription factors SOX10 (S), OLIG2 (O), and NKX6.2 (N). Here, we applied quantitative image analysis and single-cell transcriptomics to compare different transcription factor (TF) combinations for their efficacy towards robust OL lineage conversion. Compared with S alone, the combination of SON increases the number of iOLs and generates iOLs with a more complex morphology and higher expression levels of myelin-marker genes. RNA velocity analysis of individual cells reveals that S generates a population of oligodendrocyte-precursor cells (OPCs) that appear to be more immature than those generated by SON and to display distinct molecular properties. Our work highlights that TFs for generating iOPCs or iOLs should be chosen depending on the intended application or research question, and that SON might be beneficial to study more mature iOLs while S might be better suited to investigate iOPC biology.

## 1. Introduction

Myelinating oligodendrocytes (OLs) are essential for saltatory nerve conduction in the central nervous system and are involved in the metabolic support of neurons and modulation of neuronal excitability [1,2]. OLs are derived from oligodendrocyte precursor cells (OPCs) and both cell types are considered heterogeneous populations with regional specifications and functionally different states [3,4,5,6]. Moreover, OL dysfunction is associated with several major neurological diseases, e.g., multiple sclerosis, leukodystrophy, stroke, and schizophrenia [1,7]. The advent of human-induced pluripotent stem cells (hiPSCs) paved the way for generating hiPSC-derived OPCs (hiPSC OPCs) and hiPSC-derived OLs (hiPSC OLs) that improved our understanding of human biology and enabled dissection of pathophysiological mechanisms, including cell-based therapies [8,9].

Initial protocols applied small molecules and peptides but took 75 to 200 days to generate hiPSC OPCs and hiPSC OLs [8,9], thus limiting larger-scaled studies such as compound screens or large cohort investigations [10]. A subsequent, more rapid strategy used overexpression of lineage-specific transcription factors (TFs), which forces oligodendroglial differentiation and allows generation of induced OPCs (iOPCs) and iOLs within 20 to 30 days [8,9]. Most protocols identified overexpression of SRY-box 10 (SOX10) as sufficient for iOL generation [11,12,13] and overexpression of SOX9 alone was also recently shown to be an efficient approach [14]. Garcia-León et al., suggested that SOX10 (referred to as S) alone is most efficient for iOL generation [13]. In contrast, Ehrlich et al., showed that the combination SON–consisting of S, oligodendrocyte transcription factor 2 (OLIG2, referred to as O), and the NK6 homeobox 2 (NKX6.2, referred to as N)–enriched the yield of iOLs [11] and allowed direct reprogramming of human fibroblasts [15]. In addition, Pawlowski et al., applied the combination SO to generate iOLs from embryonic stem cells [12]. Nonetheless, iOLs displayed transcriptional similarities to primary OLs and allowed axonal myelination in vitro and in vivo [11,13]. Despite these functional and morphological similarities between studies, several experimental conditions beyond the different TF combinations varied, preventing direct comparison of the findings. First, the neural patterning strategies applied before initiating TF overexpression differed: Garcia et al., performed neural induction on a monolayer [13], whereas Ehrlich et al., performed free-floating embryoid body formation with neural patterning and applied different combinations of small molecules [11]. Second, different configurations of the TF-expressing lentivirus constructs were applied: Ehrlich et al., expressed all TFs as multicistronic units, however, Garcia et al., used combinations of individual TFs expressing lentiviral constructs, which likely reduced the efficacy of multi-TF combinations in generating iOLs compared with that of single TFs. Moreover, to date, no studies have addressed the fate and heterogeneity of individual iOPCs and iOL populations.

Therefore, we systematically compared S-, SO- and SON-directed differentiation of individual oligodendroglial lineage cells by using a streamlined protocol in which all TF combinations were expressed from an identical lentivirus backbone and all cell culture conditions were standardized. We show that S-, SO- and SON-directed differentiation were all sufficient to generate high yields of O4^+^ iOPCs, however, SON provided significant more yield than SO and S. Further investigations with S and SON show that SON allows an earlier generation of MBP^+^ iOLs with higher yields and more complex morphology. Subsequent scRNAseq experiments including RNA velocity analysis reveals a fastened directed oligodendroglial differentiation using SON and higher maturation stages of SON-iOLs compared to S-iOLs. We show that scRNAseq including RNA analysis is not limited to dissecting a static stage but allows to dissect the time-dependent dynamics of directed differentiation and highlights that SON-directed differentiation might be better suited for research with a focus on more mature iOL.

## 2. Materials and Methods

### 2.1. Lentiviral Vectors

The cDNA sequences of human SOX10 (S), SOX10-P2A-OLIG2 (SO) and SOX10-P2A-OLIG2-T2A-NKX6.2 (SON) were synthesized as plasmids (GenScript, Piscataway, NJ, USA), with each open reading frame flanked by attB1 and attB2 sites for Gateway recombination cloning. Synthesized genes were recombined into pDONR/Zeo (Thermo Fisher Scientific, Waltham, MA, USA, #12535035) to yield Entry clones. Entry clones were finally recombined into pINDUCER21-puro_Gateway-3xFLAG (Addgene, Watertown, MA, USA, plasmid #172981). HEK293 cells (ATCC) were used for lentivirus production. S-, SO- and SON-lentiviral vectors (12 µg of each) were transfected with packaging plasmids psPAX2 (Addgene plasmid #12260, 9 µg) and pMD2.G (Addgene plasmid #12259, 4 µg) with 4 µg polyethylenimine per 1 µg plasmid-DNA (Polysciences, Warrington, FL, USA, #9002-98-6). Then, 48 h after lentiviral transfection, the culture medium was collected, filtered through a 0.45 µM PVDF filter, precipitated with PEG-it (System Biosciences, Palo Alto, CA, USA, #LV810A-1) according to the manufacturer’s instructions, resuspended in DPBS, and stored at −80 °C for further use.

### 2.2. HiPSC Lines, HiPSC Cultivation and Lentiviral Transfection

hiPSC were derived from the hiPSC biobank at the Department of Psychiatry and Psychotherapy, University Hospital, LMU Munich, Munich, Germany. hiPSC were generated from PBMCs according to a previous protocol [16]. Conventional hiPSC verification included validation of pluripotency by immunocytochemistry (Tra1-60, NANOG, OCT4, SOX2), genomic integrity by digital karyotyping [17] (pipeline available at https://gitlab.mpcdf.mpg.de/luciat/cnv_detection.git, last accessed 28 June 2021), and successful differentiation into all three germ layers [18]. hiPSC were tested negative for HIV, HCV, CMV (by Synlab, Munich, Germany) and free of Mycoplasma infections (by Eurofins, Ebersberg, Germany). All used hiPSC lines and passaging numbers are annotated in Appendix A. Technical experiments were performed with line PSYLMUi001-A, SON-directed oligodendroglial differentiation efficiency was replicated with six independent hiPSC lines as annotated in Appendix A.

hiPSCs were cultivated in feeder-free conditions with iPS-Brew (Miltenyi Biotec, Bergisch Gladbach, Germany, #130-104-368) on Vitronectin (Thermo Fisher Scientific, Waltham, MA, USA, #A14700). Lentiviral transfection was performed in iPS-Brew, supplemented with 1 µM Y-27632 (Rock-Inhibitor, Selleckchem, Houston, USA, #S 1049) and plated with 5 × 10^4^ cells/cm^2^. Selection with 1 ug/mL puromycin (Thermo Fisher Scientific, Waltham, MA, USA, #A1113803) was performed 48 h after transfection. We performed the first passage after transfection with Accutase (Sigma-Aldrich, St. Louis, USA, #A6964) and single cells. Subsequent passaging for expansion or maintenance was performed as hiPSC clump passaging with 0.5 µM EDTA (Thermo Fisher Scientific, Waltham, MA, USA, #15575-020). An overview of used materials is annotated in Appendix A. Independent experiments were based on independent lentiviral infections.

### 2.3. Neural Induction

Neural induction and oligodendroglial differentiation were performed according to published protocols [13,19] with the modification that lentiviral transduction was performed before and not after neural induction (Figure 1A). At least one passage before neural induction, the hiPSC medium was changed to mTeSR1 (StemCell, Vancouver, Canada, #85850). Two days before neural induction, hiPSCs were singularized with Accutase, resuspended in mTeSR1, supplemented with 1× RevitaCell (Thermo Fisher Scientific, Waltham, MA, USA, #A2644501), and plated at a cell density of 2 × 10^4^ cells/cm^2^ on 12- and 24-well plates (Corning, New York, NJ, USA, #CORN3513 and #CORN3526) coated with Matrigel (BD Bioscience, San Jose, CA, USA, #354277). Two days after cultivation in mTESR1, neural induction was initiated by medium exchange to N2B27, which consists of DMEM/F-12 with GlutaMAX™ (Gibco, brand of Thermo Fisher Scientific, Waltham, MA, USA, #31331028), 1× N2 (Gibco, #17502048), 1× NEAA (Gibco, #11140035), 50 µM Mercaptoethanol (Gibco, #21985023) and 25 µg/mL Insulin (Sigma, #I9278), supplemented with 10 µM SB431542 (StemCell, Vancouver, Canada, #72232), 1 µM LDN193189 (StemCell, Vancouver, Canada, #72147) and 0.1 µM retinoic acid (RA)(Sigma-Aldrich, St. Louis, MO, USA, #R2625-50MG). A daily full media change of 0.5 mL per 24 wells and 1 mL per 12 wells was performed and media volume was doubled after day 4 of neural induction. From day 8 after neural induction, daily media change was performed with N2B27 supplemented with 0.1 µM RA and 1 µM SAG (Millipore, Burlington, MA, USA, #566660) until day 12.

### 2.4. Oligodendroglial Differentiation

Twelve- or twenty-four-well plates were coated with PLO/Laminin. First, 50 µg/mL poly-L-ornithine (Sigma, P4957) in DPBS (ThermoFisher Scientific, Waltham, MA, USA, #1419009400) was coated overnight at 37 °C. Then, 3× washing steps were performed with DPBS, and plates were incubated overnight with 10 µg/mL mouse laminin (Sigma, #L2020) at 37 °C. Day 12 NPCs were passaged as single cells upon Accutase treatment and were seeded at a density of 150,000 cells/cm^2^ in N2B27 supplemented with 0.1 µM RA, 1 µM SAG (Millipore, #566660), and 1× RevitaCell. The next day, directed OL differentiation was initiated by adding OL differentiation medium (OL-DM), which consists of N2B27 supplemented with 10 ng/mL PDGF-AA (PeproTech, brand of Thermo Fisher Scientific, Waltham, MA, USA, #100-13A), 10 ng/mL IGF1 (PeproTech, #100-11), 5 ng/mL HGF (PeproTech, #100-39), 10 ng/mL NT3 (Peprotech, #AF-450-03), 0.1 ng/mL Biotin (Sigma, #B4639), 1 mM dbcAMP (Sigma, #D0627), 60 ng/mL T3 (Sigma, #T6397) and 1 µg/mL doxycycline (Clontech, brand of Takara Bio, Shiga, Japan, #NC0424034). Full media change was performed every second day and puromycin selection was performed from Day + 2 to Day + 4 by supplementing media with 1 µg puromycin (ThermoFisher Scientific, Waltham, MA, USA, #A1113803). Cells were passaged at Day + 10 or cryopreserved for further use. Singularized cells were resuspended in OL-DM, supplemented with 1× RevitaCell and mixed 1:1 with cooled ProFreeze CDM (Lonza, Basel, Switzerland, #BEBP12-769E) for freezing purposes. Cells were stored overnight in a freezing container (Nalgene Mr. Frosty) at −80 °C and subsequently placed in liquid nitrogen.

### 2.5. O4 Microbeads Purification

To select O4^+^ cells, we performed microbead purification with anti-O4 microbeads (Miltenyi Biotec, #130-096-670) according to the manufacturer’s instructions.

### 2.6. Immunofluorescence Staining and Imaging Analysis

Before fixation, cells were washed with PBS. Then, they were fixed with 4% PFA for 10 min at room temperature (RT). Subsequently, 3× washing with PBS was performed. Permeabilization and blocking were performed with 0.1% Triton and 5% goat serum in PBS for 60 min at RT. Triton was omitted for O4 staining. Primary antibodies were added in PBS supplemented with 0.1% Triton and 1% goat serum and incubated overnight at 4 °C. On the next day, the primary antibody solution was removed by 3× PBS washing. The respective secondary antibodies in PBS were added and the cells were incubated for 60 min at RT. The secondary antibody solution was removed by 3× PBS washing and nuclei were counterstained with DAPI, which was added during the second washing step. Finally, samples were washed 1× with water and mounted onto microscope slides. Imaging was performed on an Axio Observer.Z1 (Zeiss) inverted microscope and fields of view (FOV) were taken at randomly defined but fixed positions for each well at 20× magnification. Image analysis was performed with the Fiji [20] and its plugin Simple Neurite Tracer [21]. Average values per FOV were used for subsequent statistical analysis.

### 2.7. Single-Cell RNA Sequencing and Data Analysis

#### 2.7.1. Sample Processing

Cells were detached and singularized with Accutase supplemented with DNAse1 (Merck, Darmstadt, Germany, #DN25-1G), cell aggregates were removed with 30 µm Pre-Separation Filters (Miltenyi Biotec, Bergisch-Gladbach, Germany #130-041-407), and cells were kept on ice in cold PBS with 1% BSA. Small aliquots of the samples were stained with trypan blue and cell density, viability, and multiplet rate were determined in an improved Neumann counting chamber. The density was adjusted to 2500 cells/µL. The multiplet rate was less than 7% in all samples.

#### 2.7.2. Library Preparation

The single-cell libraries were prepared with the SureCell WTA 3′ Kit (Illumina, San Diego, CA, USA) on a ddSEQ Single-Cell Isolator (Bio-Rad, Hercules, USA) in accordance with the manufacturer’s protocol. Library quality and yield were evaluated with a high-sensitivity DNA kit on a Bioanalyzer 2100 (Agilent, Santa Clara, CA, USA) according to the manufacturer’s specifications.

#### 2.7.3. Sequencing

Libraries for scRNAseq experiments were sequenced on a NextSeq550 (Illumina) in paired-end mode with 69 cycles in Read1 and 80 cycles in Read2. Subsequently, the sequencing data were demultiplexed with bcl2fastq (Illumina).

#### 2.7.4. Upstream Analysis of scRNAseq Data

Reads were quality-checked with FastQC (v0.11.9). Barcodes in Read1 were identified, added to the Read2 BAM file as an XC tag, and quality-checked with the ddSeeker tool [22], which found that 91% of barcodes were good quality in all samples. The tagged BAM files were further processed with Drop-seq tools (v2.3.0). A metabundle was created with the Ensembl annotation release 100 of the GRCh38 human genome (*create_Drop-seq_reference_metadata*). A non-human fragment (3′LTR and WPRE) and the 3xFLAG-Tag region of the expression constructs were added to the genome FASTA and GTF to determine the construct expression levels. The data were filtered for rejected barcodes. The reads were sorted by query name, and the BAM files were converted back to FASTQ for mapping. Read2 was mapped by using the STAR aligner (v2.7.5) with default settings. Aligned data were merged with the unaligned but barcode-tagged data. Functional annotation was added and any remaining substitution and synthesis errors in the cell barcodes were repaired to create the final BAM file. Digital gene expression data were extracted (*Digital Expression*) for the 2000 most abundant cell barcodes in each sample. The threshold was chosen to be well beyond the cut-off suggested by the knee plot generated by ddSeeker’s *make_graphs.R*.

In the RNA velocity analysis pipeline, to acquire separate count matrices for spliced and unspliced reads we estimated counts with DropEst (v0.8.6) instead of Drop-seq’s *Digital Expression*. The following command was used, in accordance with the suggestions of the developers of Velocyto [23]: *dropest -m -V -b -f -g%GTF file from Drop-seq metabundle% -L eiEIBA -m -c %FILEPATH%/dropEst/configs/drop_seq_velocyto.xml.* Note that the barcode tag was changed from *BC* to *XC* in *drop_seq_velocyto.xml* to make the Drop-seq output compatible with Dropest.

#### 2.7.5. Downstream Analysis of scRNAseq Data

The scRNAseq data were further processed with the R (v4.0.4) package *Seurat* (v4.0.1) [24]. The UMI count tables were imported in R, and Seurat objects were created for each sample. We filtered for features expressed in at least three cells in each dataset and for cells with at least 200 features in the first step. Mitochondrial RNA content was determined by grepping for “^*MT-”* (*PercentageFeatureSet*). We filtered the cells further for barcodes with less than 5% mitochondrial RNA content, at least 1700 UMI counts in the NPC and SON sample and 1000 counts in the S sample, and a minimum of 1000 detected features. After these filtering steps, 1194 cells were left in total.

#### 2.7.6. Normalization of scRNAseq Data

All datasets were merged into one object. To check for artifacts potentially introduced by merging, we additionally performed all analyses on each sample dataset individually. The data were normalized by the single-cell transform procedure (*SCTransform*) [25], which regressed out the percentage of mitochondrial transcripts.

#### 2.7.7. Dimension Reduction and Clustering of scRNAseq Data

Dimension reduction was performed as a principal component analysis (*RunPCA*). The previously identified 3000 most variable features (*FindVariableFeatures*) were used as input. The first 30 principal components were computed. Based on an elbow plot, as suggested in Macosko et al., 2015, we decided to use the first 15 principal components for downstream analysis. We performed SNN graph-based clustering with a resolution of 0.23 (*FindNeighbors, FindClusters*); this clustering returned 5 distinct clusters of cells. Next, we computed a UMAP embedding (*RunUMAP;* wrapper function for integration of the *Python* library *umap-learn*; [26]). Later in the project, we revisited the clustering and increased the resolution to 0.28, which split the iOPC cluster into two subclusters, which we then used for differential expression analysis.

#### 2.7.8. Integration with Bulk RNAseq Data

To integrate our data with publicly available bulk RNA-seq data, we followed the procedure from Ng et al. [14]. We used a primary cell dataset from mouse brain tissue because human datasets are commonly based on postmortem samples. The dataset is available on the Gene Expression Omnibus website under the accession number GSE52564 [27]. Briefly, we created an average expression table and excluded features that were expressed in less than 10% of all cells in our dataset. The filtering step was introduced to avoid artifacts resulting from bad coverage inherent to single-cell RNA-seq data and low expression. The resulting list was then filtered for variably expressed genes in our dataset; such genes were computed as differential expression with a minimum log2-fold change of 0.25 (*FindAllMarkers)*. This step improved the signal-to-noise ratio and reduced potential batch effects. Next, we filtered the reference dataset and our dataset for mouse/human homologs. Finally, a list of about 3000 features was entered into the analysis. From our datasets, we excluded all iOPC cells in the NI Day + 0 sample because of their low number and their high probability of being misclustered NPCs. From the reference dataset neurons, we retained myelinating oligodendrocytes (MO), newly formed oligodendrocytes (NFO), and oligodendrocyte progenitor cells (OPC). First, we calculated feature variability (*rowVars*) with the *DESeq2 R* package [28] for variance stabilization (*VarianceStabilizingTransformation;* vst). Then, we used the 2000 most variable features for the downstream analysis. A principal component analysis (PCA) was performed to check for batch effects and other artifacts, and Manhattan distances were calculated between samples and features. These distances were used for hierarchical clustering and shown in heatmaps. Additionally, Spearman correlation coefficients were calculated to ensure that our data correlated appropriately with the reference data.

#### 2.7.9. Differential Expression Analysis

Differential gene expression was analyzed with *Seurat v3* [29]. Genes were selected for expression in a minimum of 10% of all cells included in the analysis and for variability between groups. The data used were normalized but not integrated because the integration procedure smoothens expression differences, which leads to the underestimation of transcriptional differences. The results were tested in a Wilcoxon rank-sum test, and *p* values were adjusted with the Bonferroni procedure. Marker genes for characterization of the clusters were identified with a minimum log2-fold change in average expression (avg_log2FC) of 0.25 between the respective cluster and background (*FindAllMarkers()*). In pairwise comparisons between clusters, an avg_log2FC threshold of 1 was chosen because the lower threshold used for marker identification produced a high level of noise in downstream analyses.

#### 2.7.10. Analysis of Maturation Marker Load and Construct Expression

Construct counts based on the 3′LTR-WPRE sequence were included in the *sctransform* regression model for normalization (see Section 2.7.6.) and used for construct expression equivalent. The FLAG tag region was not detected in the sequencing data.

For the maturation marker load, a set of genes with increasing levels of expression during oligodendrocyte-lineage differentiation [9] and that were detected with scRNAseq was chosen, specifically: *CNP*, *CD9*, *CLDN11*, *GALC*, *PLP1*, *MAG*, *MAL*, *MBP*, *MOBP*, and *MYRF*. The normalized expression level of these genes was then summarized for each cell. A Spearman correlation factor between construct expression level and marker load was computed and tested for S and SON Day + 15 samples.

#### 2.7.11. Hypergeometric Gene Ontology Term Enrichment Analysis

For hypergeometric enrichment analyses of gene ontology (GO) terms, we used the GOrilla online tool [30]. Unranked gene lists were compared, of which one comprised all genes that entered the respective differential expression analysis (background) and the other identified differentially expressed genes (target). Only process terms went into the analysis. Enriched terms were filtered with an FDR-adjusted q-value threshold of 10^−3^ and a B value of less than 500.

#### 2.7.12. RNA Velocity Analysis

For RNA velocity analysis, we first created a loom file for each sample from the Dropest output by using the command line *Python* implementation of the *velocyto* (v0.17.17) [23]. Next, we combined these loom files for downstream analysis. Additionally, we analyzed each sample independently to make sure that merging the samples had not introduced major artifacts. We used the *scVelo Python* library [31] for the rest of the pipeline. We imported the combined loom file as anndata with the *anndata* library [32] and imported the cellIDs, UMAP coordinates, and cluster assignment from the *Seurat* pipeline. The cells were filtered for these IDs and their UMAP coordinates, and clusters were assigned. Further, the cells were filtered and normalized by using the default parameters with *pp.filter_and_normalize()*. Briefly, highly variable genes were selected, and the data were normalized by total library size and subsequently transformed to a logarithmic scale. For quality control, the fraction of unspliced transcripts was determined and found to be 13%, which is well within the lower range of 10% to 25% that is typically found in scRNAseq data. A fraction of this size is to be expected from a 3′ UTR-enriched RNA-seq dataset and is sufficient for RNA velocity estimation [31,33]. Moments were calculated and a full dynamic model was fitted to the data [31] (*tl.recover_dynamics(), tl.velocity(mode = “dynamical”*). This model was used to compute the velocity graph (*tl.velocity_graph()*).

#### 2.7.13. Latent Time and PAGA Analysis

In addition, the gene-shared latent time was computed (*tl.latent_time()*) to create a pseudo-timeline that was used to inform cell-to-cell transition inference. A PAGA analysis was computed to find cluster-to-cluster transitions (*tl.paga_plot()*) [34]. Latent times were reimported in R to be used as a measure for maturity and were tested between samples with a Wilcoxon rank-sum test.

### 2.8. Statistics

Data were first tested for normal distribution with a Shapiro–Wilk test. We used both parametric (Student’s *t*-test, ANOVA) and nonparametric analyses (two-sample Wilcoxon’s rank-sum test, Kruskal–Wallis test, chi-squared test, Fisher’s exact test), depending on the specific distribution properties of the data. Significance was set at * *p* < 0.05, ** *p* < 0.01, *** *p* < 0.001 and **** *p* < 0.0001.

## 3. Results

### 3.1. Generation of Stable S-/SO-/SON-Expressing hiPSCs and Neural Induction

To investigate TF-based differences of directed differentiation of iOPCs and iOLs with S [13], SO [12] and SON [11], we applied these TF combinations in accordance with the neural induction and differentiation protocol adapted from Garcia-Leon et al. [13,19] (Figure 1A). To allow a direct comparison, we cloned the three TFs combinations of S, SO, and SON into the same lentiviral backbone containing constitutive expression units for the reversed tetracycline transactivator (rtTA) [35] and a puromycin selection cassette (see Methods for details) (Figure 1B and Appendix A). The human coding regions of *SOX10**,*
*OLIG2*, and *NKX6.2**,* which were linked by the self-cleavage sites P2A and T2A, were under the control of the doxycycline-inducible operator (tetO) harboring a C-terminal FLAG-tag (Figure 1B and Appendix A). We demonstrated the functionality of the constructs 48 h after doxycycline induction by examining infected hiPSCs with immunostainings, which showed higher numbers of FLAG^+^ cells with S than with SO and SON (Figure 1C,D), likely reflecting the size of the different constructs and reduced protein expression upon P2A- and T2A-mediated cleavage. According to our protocol (Figure 1A), infection was performed on the hiPSC level; subsequent puromycin selection was performed to enrich for S-, SO-, or SON-hiPSC and followed by neural induction to generate PAX6^+^ and OLIG2^+^ neural precursor cells (NPCs) (Appendix A).

### 3.2. Graded Efficiency to Generate iOPCs by S-, SO- and SON-Directed Differentiation

We next tested whether S-, SO- and SON-directed differentiation differs in their potential to generate O4-positive (O4^+^) iOPCs as assessed by immunocytochemistry (ICC). All TF combinations generated high yields of O4^+^ cells + 4 and + 6 days after induction of oligodendroglial differentiation. SON provided higher yields than SO and S at both time points (Figure 2A–D). At day + 4, 38.1% ± 14.6% of all cells stained positive for O4^+^ with SON-directed differentiation, compared with 33.9% ± 13.3% with SO-directed and 28.6% ± 8.9% with S-directed differentiation (Figure 2B). At day + 6, 83.9% ± 14.6% of SON-directed, 74.4% ± 7.7% of SO-directed and 73.6% ± 8.1% of S-directed cells were O4^+^ (Figure 2D). These results clearly show a graded potential (SON > SO > S) to generate oligodendroglial lineage cells.

### 3.3. SON-Directed Differentiation Is Robust and Generates Mature iOLs More Efficiently Than S

To validate the robustness of our approach in generating O4^+^ cells, we applied SON-directed differentiation from six independently generated iPSC lines from different donors (Appendix A). The analysis revealed comparable efficiencies for all samples, and we could highlight that an additional intermediate puromycin selection from day + 2 until day + 4 enriches the yield of O4^+^ cells without the need for a mechanical purification step (Appendix A).

Given that the greatest differences were detected between SON and S, we compared these conditions in the subsequent analyses. Thus, we investigated the efficiency of SON- and S-directed differentiation in their potential to generate mature iOLs expressing the late-stage marker myelin basic protein (MBP). Compared with S-directed differentiation, SON-directed differentiation forced an earlier generation of MBP^+^ iOLs (at day + 6) and yielded higher numbers at day + 14 (37.5% ± 14.7% vs. 26.0% ± 17.0%) (Figure 2E,F). SON generated iOLs that were highly significantly larger and had a more complex ramified OL morphology (Figure 2G–J).

### 3.4. Individual Transcriptomic Profiles Reveal a Higher Level of Maturation of iOLs upon SON Differentiation

To reveal the transcriptional architecture of individual induced cells, we applied single-cell RNA sequencing (scRNAseq) after completion of neural induction (NI Day + 0) and 15 days after S (S Day + 15) and SON (SON Day + 15) induction (Figure 1A). All samples were analyzed individually (NI Day + 0: *n* = 217 cells, S Day + 15: *n* = 577 cells, SON Day + 15: *n* = 400 cells) and for analyses purposes all 1194 cells were fused together to enable direct comparisons. Shared nearest neighbor (SNN) graph-based clustering identified separate populations of NPCs and neuronal cells, one iOPC cluster, and two distinct iOL clusters, iOL1 and iOL2 (Figure 3A,B). Subsequent statistical analysis revealed a highly significant (*p* < 2.2 × 10^−16^) cell cluster composition between the baseline sample NI Day + 0 (*n* = 217 cells) compared to differentiated samples S Day + 15 (*n* = 577 cells) and SON Day + 15 (*n* = 400 cells) as well as between the differentiated samples (Figure 3C, Appendix A). This underlines the effect of the directed oligodendroglial differentiation per se as well as the impact on cell cluster composition when different combinations of directing transcription factors were applied (Figure 3A–C, Appendix A).

The most abundant cell types in the baseline neural induction Day + 0 sample were NPCs (89.4%) and neuronal cells (9.2%); a few cells were assigned to the iOPC cluster (1.4%) but no iOLs were detected (Figure 3A–C). The S-directed culture was dominated by the iOPC cluster (69.3%), and the proportion of iOL1 (3.3%) and iOL2 (5.2%) cells was much lower. In contrast, the sample from the SON-directed differentiation comprised a high fraction of iOL1 (38.5%) and iOL2 cells (16.8%), representing a significant (*p* < 0.001) 11- and 3-fold enrichment, respectively, over the S condition (Figure 3C). In contrast, the iOPC cluster was larger in S- compared to SON-directed oligodendroglial differentiation (*p* < 0.001; Figure 3C, Appendix A).

Next, we investigated the effect of S- versus SON-directed differentiation on the maturation state across the corresponding cell clusters. When plotting the cumulative OL-lineage marker load as a step function of the marker load rank, we observed that the SON-directed populations exhibited a different distribution of OL-lineage marker load (Kolmogorow-Smirnow-Test, *p*-value < 0.001) with generally higher load in SON-expressing cells (Wilcoxon’s rank-sum test, W = 44916, *p* < 0.001, Figure 3D). Of note, in both directed differentiation approaches OL-Lineage marker load was significantly associated with the lentiviral construct expression (*p* < 0.001, Figure 3E). The correlation was higher in SON-directed differentiation (R^2^ = 0.36) compared to S (R^2^ = 0.15) indicating a stronger oligodendroglial determination using SON-directed differentiation (Figure 3E).

In both S- and SON-directed oligodendroglial differentiation, a small fraction of cells escaped the oligodendroglial lineage and differentiated towards a neuronal identity. However, this neuronal population was 2.5 times (*p* < 0.001) less abundant with SON-directed (8.8%) than with S-directed differentiation (22.2%) (Figure 3A–C).

### 3.5. Co-Clustering of S- and SON-Differentiated iOPCs and iOLs with Primary OPCs and OLs

To compare iOPCs and iOLs with primary OPCs and OLs, we calculated the average gene expression in each cluster and sample (Appendix A) and performed unsupervised clustering of each sample’s clusters with reference transcriptomic data that were obtained from FACS purified primary cells [27]. In the resulting dendrogram, we found three main clusters: One cluster grouped the iOL1 and iOL2 profiles together with newly formedmurine oligodendrocytes (NFO) and myelinating oligodendrocytes (MO) [27], a second cluster contained the iOPCs and NPCs together with the reference OPC samples (Figure 3F) and in the third cluster the neuronal cells clustered together with primary neurons. When we looked at the 50 most variable features in this unsupervised clustering, we found that 26 of those genes formed a gene set that separated the oligodendroglial samples from the neuronal cells (Appendix A). In this gene set, all iOL cluster profiles closely resembled the NFO and the expression profile MO of the reference sample. In a principal component analysis, the first principal component explained 48% of the total variance, and the samples were aligned in the presumable flow of differentiation, findings that underline the high similarity between the S-/SON-directed hiPSC-derived cells and primary oligodendroglial cells (Figure 3G).

### 3.6. Dedicated Gene Expression Profiles of Marker Genes Support Identities of Subclusters and Differential Efficiency of S and SON Conditions towards the Oligodendroglial Lineage

To further characterize cluster identities, we compared the expression and abundance of marker genes [9] in the dedicated NPC, iOPC, and iOL1 and iOL2 subclusters (Figure 4 and Appendix A, Appendix A). The neural stem cell marker *PAX6* was highly expressed not only in a large proportion of NPCs but also in a substantial proportion of iOPCs. S-derived iOPCs expressed *PAX6* at higher frequencies as compared to SON iOPCs (*p* < 0.001) (Figure 4A–C). This enrichment towards NPC/S-iOPC was supported by violin plots showing the distribution of the averaged expression levels which was higher in S iOPCs over SON iOPCs (*p* < 0.001, Figure 4D). An overall similar profile was observed for an additional progenitor marker, *SOX3* (Appendix A, Appendix A). *SOX9*, an OPC stage-enriched TF, was predominantly expressed in S- and SON-iOPCs with no substantial differences within the respective clusters but a significantly higher number of *SOX9* expressing cells in the S-directed cluster compared to the SON condition (*p* < 0.001, Figure 4E–H). An overall similar, OPC-biased profile was obtained with *ST8SIA1* (the gene product stained with the prototypical OPC marker A2B5, Appendix A, Appendix A). *MBP* was selected as a marker for mature OLs and showed a strong increase in expression comparing iOPC and iOL1 clusters (*p* < 0.001) reaching the highest expressing levels in the iOL2 cluster (*p* < 0.001, Appendix A). Moreover, SON iOPCs express low levels of *MBP* in contrast to S iOPCs *(p* < 0.001), but in S-derived cells, *MBP* was almost exclusively detected in iOL1 and iOL2 clusters (Figure 4I–L). Notably, some myelin genes expressed at the late stage, such as myelin-associated oligodendrocyte basic protein (*MOBP*), were robustly expressed only in the SON-derived iOL2 cell population and almost absent in S-derived cells (Figure 4M–P, Appendix A). The higher level of expression of *MBP* and many other myelin marker genes, such as proteolipid protein 1 (*PLP1*) and galactosylceramidase (*GALC*, Appendix A, Appendix A), indicates that the iOL2 subcluster represents the most matured fraction of iOLs.

### 3.7. RNA Velocity Reveals Divergent Dynamics of Fate Decisions upon Induced Oligodendroglial Differentiation

To examine the transcriptional dynamics underlying the different processes and outcomes of S- and SON-directed differentiation, we applied the RNA velocity approach to model the splicing kinetics as a proxy of the status of individual cells along the differentiation trajectories [23]. Briefly, the direction and speed of transcriptional changes in single cells are estimated from the ratio of detected unspliced to detected spliced transcripts. This allows for the prediction of the future transcriptional state of a specific gene in a given cell. A higher than expected ratio of unspliced to spliced mRNAs signifies upregulation over steady-state expression levels because of the temporal delay between an increase in transcriptional rate and an increase in the amount of mature mRNAs as end products of splicing [23]. In turn, a lower than expected unspliced to spliced ratio indicates downregulation because of the temporal delay between a decrease in the transcriptional rate and degradation of mature mRNAs. The ‘velocity’ vectors represent the direction and speed of the transcriptional changes in each cell and estimate future cell kinetics by indicating the movement of the cell through transcriptional space during differentiation.

In our data, we first determined the proportion of detectable unspliced transcripts to ensure their suitability for this type of analysis. This value typically depends on the protocol used, data processing, read length, and cell types present in the sample [31,33]. In our samples, the overall proportion of unspliced counts was 13% (Appendix A), which was well within the expected range [33]. We found small differences among clusters (Appendix A) and between samples (Appendix A) but the differences were all within a range of 11% (iOL1) to 16% (NPC).

When the ‘velocity’ vectors were superimposed on the UMAP dimension plot (Figure 3A), two main streams of cells—an oligodendroglial and a neuronal stream—emerged along differentiation pathways in our cultures (Figure 5A). Moreover, we looked at the gene-shared latent time, a recent pseudo-time concept that uses only transcriptional kinetics instead of a diffusion-based approach [31]. Here, we found two endpoints, one pathway in the direction of the neuronal cluster and one in the direction of the iOL2 cluster, again indicating the presence of two distinct lineages in the pooled S and SON samples (Figure 5B). Additionally, we used partition-based graph abstraction (PAGA) to identify transitions between clusters [34] (Figure 5C). These transitions confirmed the two major differentiation pathways in our samples: One pathway represented the oligodendroglial lineage, in which the cells transitioned from an OPC state to an early OL identity in the iOL1 cluster and then moved on to the more mature iOL2 cluster, and the other led to a neuronal identity. In this stream, cells in the neural induction baseline sample differentiated directly towards this direction. In contrast, in the S and SON samples, where no NPCs were left, iOPCs transdifferentiated to a neuronal identity. These escaping iOPCs were also clearly visible in the vector plot (Figure 5A).

In addition, we analyzed the dynamic model and RNA velocity of individual ‘velocity genes’ (Figure 5I). An interesting example of a strong ‘velocity gene’ is *MBP*. The phase portrait visualizes unspliced and spliced transcripts overlayed with the corresponding dynamic model (Figure 5D). When examining the phase portrait of *MBP*, we noticed that most iOL1 cells were in an on state, i.e., upregulating *MBP* transcription, as indicated by a high positive ratio of unspliced to spliced transcripts (Figure 5E). In contrast, most iOL2 cells were close to a steady-state or negative velocity value, reflecting the endpoint of *MBP* upregulation (Figure 5E). However, the somewhat counterintuitive RNA velocity-derived model for the oligodendroglial differentiation path iOPC to iOL1 to iOL2 (Figure 5C,E) was in line with the elevated levels of spliced *MBP* mRNA level along this trajectory (Figure 5F,I).

We also detected clear differences in the differentiation processes between the S Day + 15 and SON Day + 15 samples (Figure 5G). Notably, the population of iOPCs moving towards the neuronal cluster was much more prominent in S-directed than in SON-directed oligodendroglial differentiation (Figure 5G, upper light blue half of the iOPC cluster). Moreover, the gene-shared latent time scores of each cell could be used as an indicator of maturity because in both samples the cells took a similar path in their differentiation to OLs. This was confirmed in an individual RNA velocity analysis of both samples, which eliminated the possibility that one sample could be interfering with the other (Appendix A). When comparing the latent time scores between samples, the oligodendroglial lineage in the SON-expressing culture (median = 0.66; MAD = 0.31; *n* = 365) was found to be significantly more mature than that in the S-expressing culture (median = 0.30; MAD = 0.05; *n* = 449) [W = 42640; *p* = 2.2 × 10^−16^] (Figure 5H). This result was in line with the higher abundance of iOL1 and iOL2 cells in SON Day + 15.

Finally, we identified a series of differentiation marker genes from a previously published study [9] expressed in our samples and illustrated their transcriptional kinetics during OL differentiation as a gene expression level dependent on latent time (Figure 5I). This allowed us to show again the high overlap of a late gene-shared latent time point, the SON sample, and the iOL cluster identities (Figure 5I).

### 3.8. Characterisation of iOPC Subclusters

The RNA velocity analysis enabled us to clearly identify two distinct differentiation streams within the iOPC cluster that were moving in different directions (Figure 5G), a feature that was not detected in the first shared nearest neighbor (SNN)-based clustering result (Figure 3A,C). One subcluster followed the oligodendroglial lineage and was present in both S- and SON-directed differentiation and another partly escaped towards a neuronal lineage and was almost exclusively found in the S-directed differentiation (Figure 5G). Because of this observation, we increased the resolution of the SNN clustering to empirically identify subpopulations in the sample. This approach allowed us to identify two iOPC subclusters that were illustrated in an RNA velocity plot color coded for the iOPC subclusters (Figure 6A and Appendix A). The iOPC2 subcluster, which occurred in both conditions (18.2% in S Day + 15 and 32.0% in SON Day + 15), comprised mainly the iOPC population of the OPC-OL differentiation pathway (Figure 6A). In contrast, the iOPC1 subcluster was almost exclusively observed in S Day + 15 (51.1% in S Day + 15 and 4.2% in SON Day + 15) and comprised the stream of cells escaping towards the neuronal cluster, which likely represented a more volatile iOPC stage under the given culture condition (Figure 6A and Appendix A). Of note, we analyzed the expression of the exogenous lentiviral construct via the scRNAseq-detecTable 3′LTR site (Appendix A). Expression levels of exogenous constructs were higher and more homogenous in iOL1 and iOL2 compared to iOPC1, iOPC2, and the neuronal cluster. Furthermore, within iOPC1, iOPC2, iOL1, and iOL2, S-, or SON-construct expression were comparable.

To identify molecular signatures of the two iOPC stages, we compared the differential average gene expression in the S-iOPC1, S-iOPC2, and SON-iOPC2 subclusters (Figure 6B–F, Appendix A). We found substantial differences in significantly deregulated genes between the iOPC1 and iOPC2 clusters in the comparisons of S-iOPC1 with S-iOPC2 (Figure 6B) and S-iOPC1 with SON-iOPC2 (Figure 6C). Notably, the progenitor markers PAX6 and SOX3 were expressed at a lower level in the iOPC2 cluster, indicating a higher maturation level (Figure 6B,C). In contrast, only very few genes were found to be differentially expressed in S-iOPC2 and SON-iOPC2, indicating that the iOPC2 cluster identity was independent of the condition and that it represents a rather homogenous entity (Figure 6D). This assumption was corroborated by the broad overlap of differentially expressed genes in both comparisons (Figure 6E,F). A comparison of SON-iOPC1 and SON-iOPC2 was deemed not advisable because of the low number of cells in the SON-iOPC1 cluster, which can be independently interpreted as an indicator of an increased drive of iOPC maturation mediated by SON compared with iOPC maturation mediated by S alone. This finding underlines that the directed OL differentiation driven by S was similar in both conditions, whereas the efficiency was higher with SON than with S. Furthermore, we performed a gene ontology term-based hypergeometric enrichment analysis for the comparisons of iOPC1 and iOPC2 (Appendix A). Thereby we identified gene-sets relevant for the organization of the extracellular matrix as most significantly altered (Figure 6G), a result that is in line with several genes of the collagen family that were detected as being upregulated in the gene-by-gene analysis (Figure 6B,C).

These findings prompted us to hypothesize that structural differences may also be visible between S- and SON-generated iOPCs. Therefore, for both conditions we performed microbead purification to enrich O4^+^ iOPCs at day + 10 and morphological analysis of O4^+^ cells at day + 16. We thereby observed substantial morphological differences between S- and SON-directed O4^+^ cells (Figure 6D) and a larger average cell size of SON-derived O4^+^ cells compared with S-derived O4^+^ cells (Figure 6H).

Overall, our results show that different combinations of TFs have a substantial effect on iOPC and iOL morphology, maturation stage, cell cluster composition, and lineage transition dynamics.

## 4. Discussion

In our comparative study, we confirmed that S-, SO- and SON-directed conversion are all sufficient to rapidly generate oligodendroglial lineage cells from hiPSCs. As a result of the side-by-side analysis at the single-cell resolution, we showed that the application of SON generated more O4^+^ late-stage iOPCs and more mature and complex MBP^+^ iOLs than the application of S alone, supporting previous observations [11]. S alone was most effective in generating a mixed population of iOPCs (see below). In agreement with previous studies [36,37], we observed a higher level of expression of smaller-sized constructs. However, an intermediate selection step performed in our protocol normalized these differences and allowed us to compare TF combinations irrespective of construct size. As mentioned above, a more recent study that used a systematic screen for lineage-converting TFs found that SOX9 was capable of inducing oligodendroglial cells on its own [14]. Thus, similar to other SOX family members, both SOX9 and SOX10 can be considered as pioneering TFs in the oligodendroglial lineage that prime critical steps essential for stem cell conversion [38]. SOX9 operates genetically upstream of SOX10 in the oligodendroglial lineage and—despite the fact that SOX9 can generate myelinating iOLs in a 3D system [14]—it remains an open question whether SOX9 might generate more earlier stage iOPCs than SOX10 in a cell-autonomous setting such as the one applied in this study. We observed substantial differences in iOPC and iOL heterogeneity and dynamics of differentiation between S- and SON-directed oligodendroglial differentiation. Despite the finding that OPC marker genes such as *SOX9*, *ST8SIA1*, *CNP*, *ID2* and *ID4* reached similar levels in S- and SON-iOPCs, the expression level of oligodendroglial maturation markers such as *GALC* and *MOBP* was higher in SON-iOLs than in S-iOLs under the applied conditions. Moreover, the cell cluster composition differed dramatically: Mature iOLs, that express *PLP1, GALC, MBP, MOBP,* were far more abundant upon SON-directed differentiation but the iOPC cluster was larger and more diverse in the S-directed condition (see below).

To improve purity and scalability, we introduced the lentivirus-encoded lineage transcription factors at the level of iPSC cultivation followed by two puromycin selection steps, one before and one after initiation of iOPC/iOL differentiation; this protocol was robust for different iPSC lines obtained from six different human donors. The advantage of this approach is that less lentivirus is needed and only infected S/O/N-hiPSCs are proliferative and can be amplified, frozen, banked, and recovered en masse at the time needed. In the future, this approach will be particularly suited to enable larger-scaled case-control comparisons and may allow also genetic and compound screenings requiring high numbers of dedicated cells. Nonetheless, epigenetic lentivirus silencing is a confounding factor [39]. The drop of FLAG^+^ hiPSCs after doxycycline induction is likely due to the loss of cleavage efficiencies by single versus double P2A/T2A element-containing constructs [40].

In both conditions (S and SON), the lentiviral construct expression level was associated with a higher level of OL-lineage marker load. However, SON-directed differentiation caused a much higher marker load underlining a more maturating oligodendroglial differentiation compared to S-directed differentiation in line with previous work [11].

On the iOPC/iOL level, lentiviral construct transcriptome levels within the clusters were comparable between S and SON indicating that the expression levels are unlikely to cause cluster diversity and that this diversity is most likely attributed to the different TF combinations. However, expression levels were higher in iOL clusters compared to iOPC clusters and the neuronal cluster. Thus, it might be possible that higher expression levels sustain the directed differentiation. Thus, for genetic and chemical screens [41,42], safe harbor strategies may be a better choice to express OL-lineage-converting TFs and previous work could highlight that safe harbor strategies using SOX10 were sufficient to generate high yields of homogenous MBP^+^ iOLs [19].

However, safe harbourstrategies will be complicated to apply in case-control comparisons, where many subjects/cell lines are to be compared. Nonetheless, a safe harbor-mediated CRISPR/Cas9-based genetic screen has been described for hiPSC-derived NGN2-directed iNeurons [41], showing the potential of this approach also for other cell types. Safe harbor strategies that apply CRISPR/Cas9 technology could also be applied to modify directed differentiation. In contrast to exogenous overexpression of TFs for directed differentiation, a recently published study applied CRISPR/Cas9-mediated activation of endogenous *Sox10*, *Olig2*, and *Nkx6.2* by specific gRNAs to differentiate mouse neural stem cells and mouse embryonic fibroblasts to O4^+^ and MBP^+^ oligodendroglial cells [43].

In this study and previous studies that used the same neuronal patterning approach [13,19], the expression of early-stage OPCs markers, such as *CSPG4* (*NG2*) and *PDGFRA*, was barely detectable in iOPCs. Thus, with both SON and S this directed differentiation approach can be assumed to bypass early OPC stages. Chemical differentiation approaches may therefore be preferable for examining early-stage OPCs over a longer time period [44,45]. A study that used such an approach applied scRNAseq on PDGFRA-enriched, chemically transformed OL-lineage cells and uncovered several stages of OPCs [46]. The scRNAseq analysis of our study revealed that directed oligodendroglial-lineage conversion also reflects several aspects of known OPC and OL development [9,47]. For example, we observed an upregulation of *ST8STIA1* (A2B5), *ID2*, *ID4*, and *SOX9* in iOPCs. In addition, we found increased expression of the oligodendroglial markers *CNP* and *PLP1* during differentiation. In addition, in more mature iOLs we found an enhanced expression of marker genes for OL differentiation, such as *MBP* and *MYRF* that are crucial for myelination [47].

To investigate hiPSC-based oligodendroglial lineage differentiation, we applied RNA velocity analysis, a technique that was developed for investigating differentiation processes in scRNAseq datasets to infer temporal changes without needing to obtain samples at several time points [23]. Thereby, we described the transcriptional dynamics during fate decisions in induced oligodendroglial differentiation and identified *MBP*, for example, as a velocity gene. Nevertheless, our data are based on 3′UTR-enriched single-cell transcriptome libraries, which limits their information content in RNA velocity analysis. Full-length scRNAseq protocols, which cover more splicing sites, would detect a higher variety of splicing variants and thus increase the number of potential velocity genes and might be beneficial to improve velocity analysis [31]. Nonetheless, the RNA velocity-based analyses in this study provided additional insights. With the help of graph-based analysis methods such as PAGA [34], we could visualize the oligodendroglial differentiation stream from iOPC to iOL1 and iOL2, which could be validated independently in S- and SON-directed differentiation and which would not have been identified with standard scRNAseq analysis. To generate a pseudo-timeline, we used gene-shared latent time instead of common similarity-based Markovian diffusion models, as suggested by Bergen et al., (2020). This approach improved the resulting timeline dramatically because the diffusion-based pseudo-timeline forced a unidirectional ‘time flow’ onto the dataset, which clearly did not fit the existence of two distinct lineages in one dataset. We used latent time analysis to highlight the ‘time’-dependent up- and downregulation of several oligodendroglial stage markers and to estimate the maturation state of a single cell within the whole span of oligodendroglial differentiation. Hereby, we observed a higher latent time distribution with SON than with S, indicating that oligodendroglial differentiation was faster in the SON- than in the S-directed differentiation. RNA velocity also uncovered a second neuronal leakage stream that was more prominent in S-directed differentiation. This leakage stream indicated that not only NPCs with low construct expression but also parts of the S-iOPC1s differentiated towards a neuronal state. Thus, we found evidence that S-directed differentiation might allow some iOPCs to ‘escape’ towards the neuronal lineage during the differentiation process, and that SON-directed differentiation seems to be less volatile and more efficient regarding OL differentiation under the applied conditions.

Differential gene expression analysis revealed that S-iOPC2 and SON-iOPC2 were quite similar and showed a substantial overlap in their differentially expressed genes when compared with iOPC1. This finding hints at a similar differentiation path with S and SON, although efficiency was higher with SON, as reflected by the abundance of more mature oligodendroglial cells, i.e., iOPC2, iOL1, and iOL2. On the other hand, iOPC1 constitutes a population of ‘escaping’ cells, giving us the opportunity to investigate the transcriptional networks and signaling machinery necessary for different steps of the differentiation process. Pathway analysis indicated greater expression of extracellular matrix components in iOPC2 cells than in iOPC1 cells; this difference may reflect a maturation step, as indicated by the change in direction of the velocity vectors towards the OL stage, or may represent a slight deflection towards other glial lineages [46].

Oligodendroglial-lineage cells represent a heterogeneous cell population with a high spatio-temporal diversity [3,6]. A limitation of directed protocols is that they most likely generate only a subset of OPC/OL sub-lineages, depending on which TFs or combination of TFs they use. An advantage of directed conversion, however, is that human cells from a defined differentiation state can be generated in virtually unlimited amounts for various applications.

Our work highlights that applied TFs for generating iOPCs or iOLs should be chosen depending on the intended application or research question. Our analysis suggests that differentiation directed by S offers a useful tool for research on OPC biology and earlier stages of oligodendroglial differentiation, including research that addresses the effects of external cues promoting OPC and OL differentiation, i.e., in a 3D/spheroid culture system; the same might be true for SOX9 [14]. On the other hand, applications, and research with a focus on more mature iOLs might benefit from the use of SON. TF combinations such as SON are likely to be better suited to studying OL biology and myelination-associated aspects, given the much higher conversion rate and easier access to much higher numbers of mature OLs with such combinations. Further investigations are certainly needed to unleash the potential of future cell therapies with engineered iOLs in myelin diseases, but a better understanding of the nature of induced cells via different TF-directed protocols will certainly be helpful to achieve this goal. We believe that the molecular and analysis tools applied in this study to force and dissect human oligodendrocyte development may support the understanding of dynamic aspects of cell-autonomous versus non-cell-autonomous contributions in in vivo disease models in the future, i.e., via grafting of iOLs generated from normal and diseased donors into mouse brains. The paradigmatic study on human iPSC-derived glia/mouse chimeras already revealed oligodendroglia-cell-autonomous contributions to childhood-onset schizophrenia [48]. More such in vivo studies are needed that combine patient-derived oligodendoglial cell types with sophisticated single-cell level analyses to further increase our knowledge of disorders with myelination deficits.

## Figures and Tables

**Figure 1 cells-11-00241-f001:**
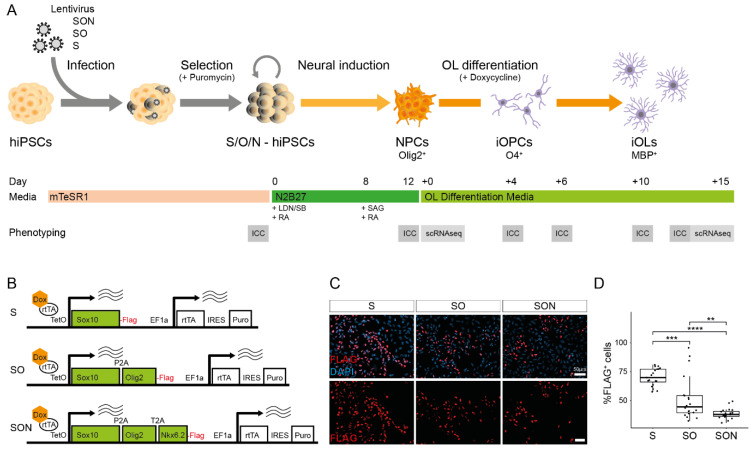
Experimental overview and applied constructs. (**A**) Experimental overview of directed differentiation of induced oligodendrocytes (iOLs) and their precursor cells (iOPC) from human induced pluripotent stem cells (hiPSCs). (**B**) Scheme of lentiviral vectors expressing the transcription factors (TF) SON (SOX10-OLIG2-NKX6.2), SO (SOX10-OLIG2) and S (SOX10). (**C**) Representative images of FLAG^+^ hiPSCs 48 h after doxycycline induction. Scale bar: 50 µm. (**D**) Quantification of the percentage of FLAG^+^ cells 48 h after doxycycline induction. Mean ± SD of FLAG^+^ cells: 70.3% ± 7.5% in S-hiPSCs, 51.6% ± 18.7% in SO-hiPSCs and 38.7% ± 4.5% in SON-hiPSCs. Illustrated with box (quartiles) and whisker (largest/smallest observation within hinge ± 1.5× interquartile range) plot. Each dot represents one analyzed field of view (*n* = 20 per condition) from one independent experiment. Statistical analysis by Kruskal–Wallis test (*p* < 0.0001) and post hoc Mann–Whitney U test. ** *p* < 0.01, *** *p* < 0.001, **** *p* < 0.0001.

**Figure 2 cells-11-00241-f002:**
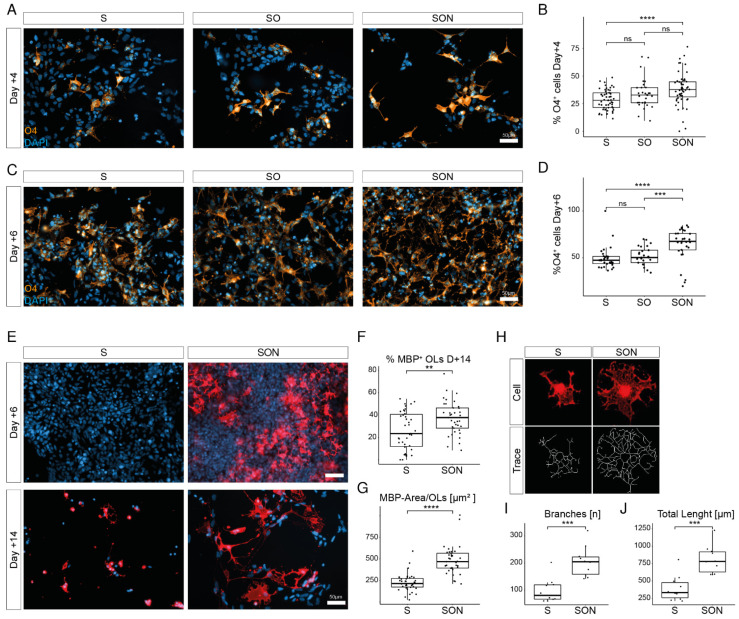
SON-directed differentiation leads to accelerated differentiation and morphologically highly ramified oligodendrocytes as compared with S-directed differentiation. (**A**) O4^+^ (orange) iOPCs at Day + 4 of oligodendrocyte differentiation with S, SO and SON. Scale bar: 50 µm. (**B**) Quantification of O4^+^ cells at Day + 4 illustrated with box-and-whisker plot. Each dot represents one analyzed field of view (*n* = 30–55 per condition) from two independent experiments. Statistical analysis by ANOVA (F = 8.02, *p* = 0.00051) and post hoc Tukey test. **** *p* < 0.001, ns = *p* > 0.05. (**C**) O4^+^ (orange) iOPCs at Day + 6 of OL differentiation with S, SO and SON TFs. Scale bar: 50 µm. (**D**) Quantification of O4^+^ cells at Day + 6 illustrated with a box-and-whisker plot. Each dot represents one analyzed field of view (*n* = 25–29 per condition) from two experiments. Statistical analysis by Kruskal–Wallis test (*p* = < 0.0001) and post hoc Mann–Whitney U test, *** *p* < 0.001, **** *p* < 0.0001, ns = *p* > 0.05. (**E**) MBP^+^ iOLs appear at Day + 6 with SON-directed differentiation; no MBP^+^ iOLs are detectable at this timepoint in the S condition. At Day + 14, SON- and S-MBP^+^ cells are morphologically different. Representative images from two independent experiments. (**F**) Quantification of MBP^+^ iOLs at Day + 14 illustrated with a box-and-whisker plot. Each dot represents one analyzed field of view (*n* = 37–39 fields of view per condition). Statistical analysis by Mann–Whitney U test. ** *p* < 0.01. Representative data from two independent experiments. (**G**) Quantification of average cell size of MBP^+^ SON-iOLs compared with S-iOLs at Day + 14 of differentiation. Box-and-whisker plot; Each dot represents one analyzed field of view (*n* = 37–39 fields of view per condition). Statistical analysis by Mann–Whitney U test. **** *p* < 0.0001. Representative data from two independent experiments. (**H**) Illustration of representative individual cell assessments of S-iOLs and SON-iOLs with the Neurite simple tracer. (**I**) Quantification of the number of branches per MBP^+^ cell (*n* = 10 analyzed cells per condition) illustrated by box-and-whisker plot. Statistical analysis by Mann–Whitney U test. *** *p* < 0.001. (**J**) Quantification of the total traced length per MBP^+^ cell (*n* = 10 analyzed cells per condition) illustrated by box-and-whisker plot. Statistical analysis by Mann–Whitney U test. *** *p* < 0.001.

**Figure 3 cells-11-00241-f003:**
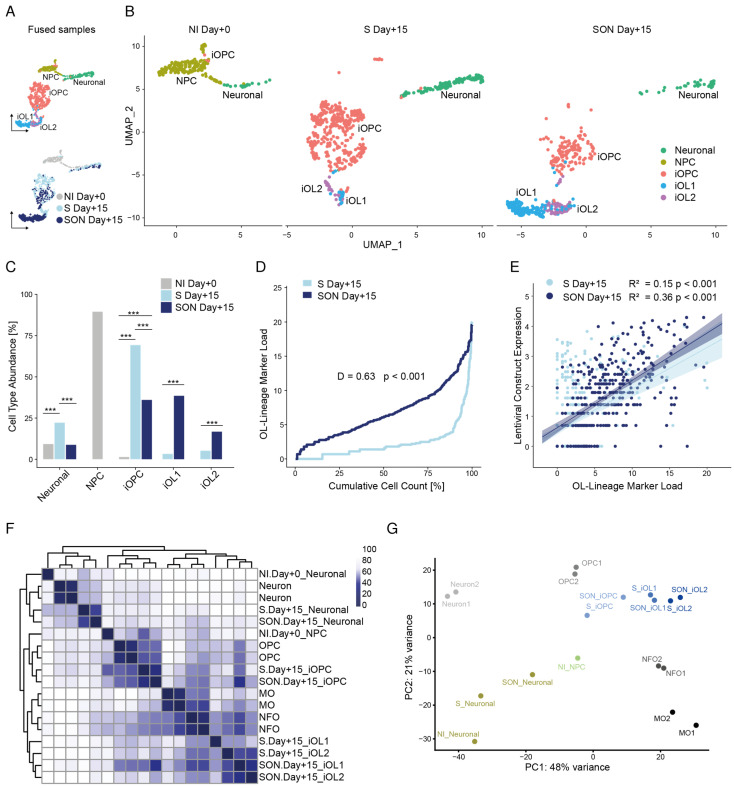
scRNAseq reveals higher level of maturation of oligodendroglial lineage cells, increased level of lentivirus dependent lineage commitment in SON-differentiation and co-clustering with primary OPCs and OLs. (**A**) UMAP embedding of integrated scRNAseq data of baseline sample (Day + 0) after neural induction (NI, *n* = 217 cells) and samples + 15 days after directed differentiation by SOX10 (S, *n* = 577 cells) and SOX10-OLIG2-NKX6.2 (SON, *n* = 400 cells). Top: UMAP dimension plot illustrates clusters of neural precursor cells (NPC, brown), neuronal cells (green), induced oligodendrocyte precursor cells (iOPCs, red), induced oligodendrocyte cluster 1 (iOL1, blue), and induced oligodendrocyte cluster 2 (iOL2, violet). Bottom: UMAP dimension plot illustrated sample composition after neural induction (NI Day + 0, grey), 15 days after directed differentiation by SOX10 (S Day + 15, light blue), and SOX10-OLIG2-NKX6.2 (SON Day + 15, dark blue). (**B**) UMAP dimension plot, split by sample, showing clusters of neural precursor cells (NPC, brown), neuronal cells (green), induced oligodendrocyte precursor cells (iOPCs, red), induced oligodendrocyte cluster 1 (iOL1, blue), and induced oligodendrocyte cluster 2 (iOL2, violet). (**C**) Percentage of cells (*y*-axis) in each cluster (*x*-axis) from scRNAseq samples NI Day + 0 (grey), S Day + 15 (light blue), SON Day + 15 (dark blue) from B. Significant differences (*** *p* < 0.001) of the cell cluster abundance between annotated samples based on Fisher’s exact tests. (**D**) Cumulative iOL lineage marker load (*y*-axis) across the ranked cells (*x*-axis) after + 15 days of directed differentiation by SOX10 (S Day + 15, light blue) and SOX10-OLIG2-NKX6.2 (SON Day + 15, dark blue). *p*-value indicates a significance difference in iOL lineage marker load distribution (Kolmogorov–Smirnov test, *p*-value < 0.001) across the cell population from the scRNAseq experiments after S- and SON-directed oligodendroglial differentiation with higher load in SON-expressing cells (Wilcoxon’s rank-sum test, W = 44916, *p* < 0.001. (**E**) Linear regression model of the relationship between lentiviral construct expression (*y*-axis) and iOL lineage marker load (*x*-axis) across the cell populations of S- and SON-directed oligodendroglial differentiation from corresponding scRNAseq data. Spearman’s R^2^ coefficient of determination and respective significance value of the regression model is indicated in the plot. (**F**) Comparison of transcriptomes of samples from this study with published primary murine cells [27]. Heatmap of Manhattan sample distances and dendrograms from the corresponding unsupervised hierarchical clustering. Abbreviations for primary reference samples: MO, myelinating oligodendrocytes; NFO, newly formed oligodendrocytes; OPC, oligodendrocyte precursor cells. (**G**) Principal component analysis dimension plot showing the first two principal components. scRNAseq data points are colored, whereas reference murine samples are shown in black and grey scale, as indicated.

**Figure 4 cells-11-00241-f004:**
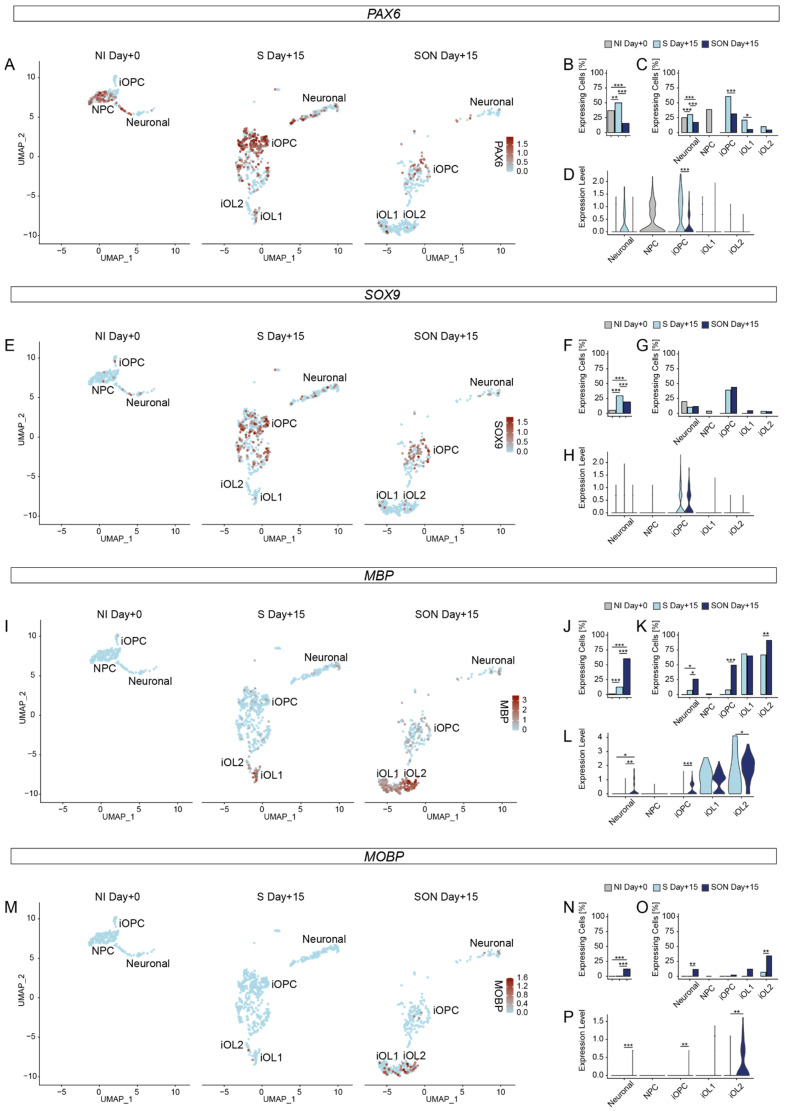
Marker gene expression verifies cluster identities and shows a mature oligodendrocyte phenotype in SON. (**A**,**E**,**I**,**M**) UMAP representation showing single-cell expression of selected marker genes (**A**) *PAX6*, (**E**) *SOX9*, (**I**) *MBP*, (**M**) *MOBP* from scRNAseq of baseline sample after neural induction (NI Day + 0, *n* = 217 cells, one independent experiment) and samples after 15 days of directed differentiation by SOX10 (S Day + 15, *n* = 577 cells, one independent experiment) and SOX10-OLIG2-NKX6.2 (SON Day + 15, *n* = 400 cells, one independent experiment) with annotated cell clusters of neural precursor cells (NPC), neuronal cells, induced oligodendrocyte precursor cells (iOPCs), induced oligodendrocyte cluster 1 (iOL1) and induced oligodendrocyte cluster 2 (iOL2). Expression value per cell plotted according to the color intensity of the respective scale bar as indicated. (**B**,**F**,**J**,**N**) Bar plots illustrate the abundance of expressing cells for each sample. ** (*p*-value < 0.01), *** (*p*-value < 0.001) based on Fisher’s exact tests (details provided in Appendix A). (**C**,**G**,**K**,**O**) Bar plots illustrate the abundance of expressing cells for each cluster split by sample. * (*p*-value < 0.05), ** (*p*-value < 0.01), *** (*p*-value < 0.001) based on Fisher’s exact tests (details provided in Appendix A). (**D**,**H**,**L**,**P**) Violin plots of the normalized expression per sample within the respective cell type cluster. * (*p*-value < 0.05), ** (*p*-value < 0.01), *** (*p*-value < 0.001) based on Wilcoxon signed-rank test (details provided in Appendix A).

**Figure 5 cells-11-00241-f005:**
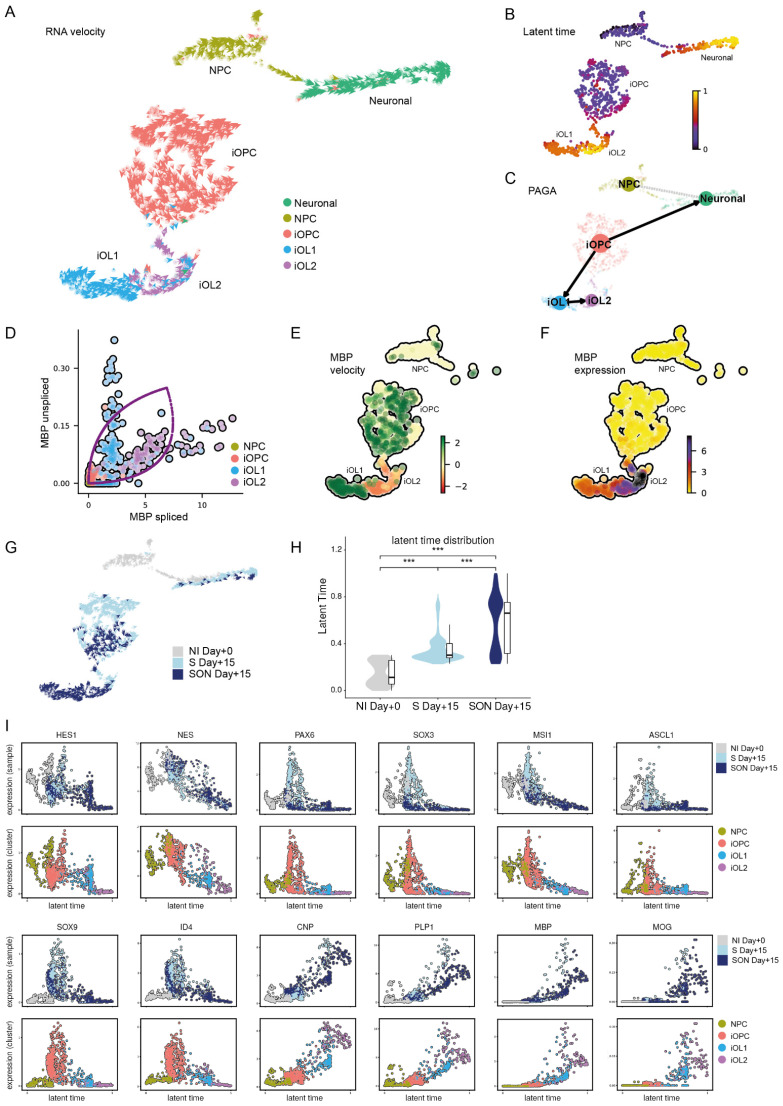
RNA velocity reveals the dynamics of fate decisions upon induced oligodendroglial differentiation. (**A**) RNA velocity vectors projected on the single-cell UMAP-based dimension plot indicate direction and speed of individual cells in transcriptional space. (**B**) The gene-shared latent time identifies two endpoints of differentiation in the scRNAseq samples: One in the induced oligodendrocyte cluster 2 (iOL2) cluster and one in the neuronal cluster. (**C**) Partition-based graph abstraction (PAGA) plot visualizing transitions between clusters superimposed on the UMAP embedding reveals two differentiation paths: (1) the oligodendroglial lineage, in which the cells transition from iOPC to iOL1 to iOL2, and (2) the neuronal lineage from NPCs and iOPCs that escape the TF-directed oligodendroglial differentiation. (**D**) Dynamic model of MBP’s transcriptional kinetics plotted on top of the scatter plot of unspliced and spliced transcript expression. (**E**) MBP velocity estimates indicating MBP mRNA induction (green) and repression (red). (**F**) MBP expression level. (**G**) Velocity vectors on UMAP embedding with sample color code: Sample after neural induction (NI Day + 0) in grey, S-directed differentiation (S Day + 15) in light blue and SON-directed differentiation (SON Day + 15) in dark blue, as indicated. (**H**) Latent time distribution between baseline (NI Day + 0), S-directed differentiation (S Day + 15), and SON-directed differentiation (SON Day + 15), illustrated with a violin and box- and-whisker plot. Pairwise comparisons were tested with Wilcoxon’s rank-sum test. *** *p* < 0.001. (**I**) Latent time versus gene expression of stage-specific genes during oligodendrocyte differentiation. Individual cells colored by cluster and sample.

**Figure 6 cells-11-00241-f006:**
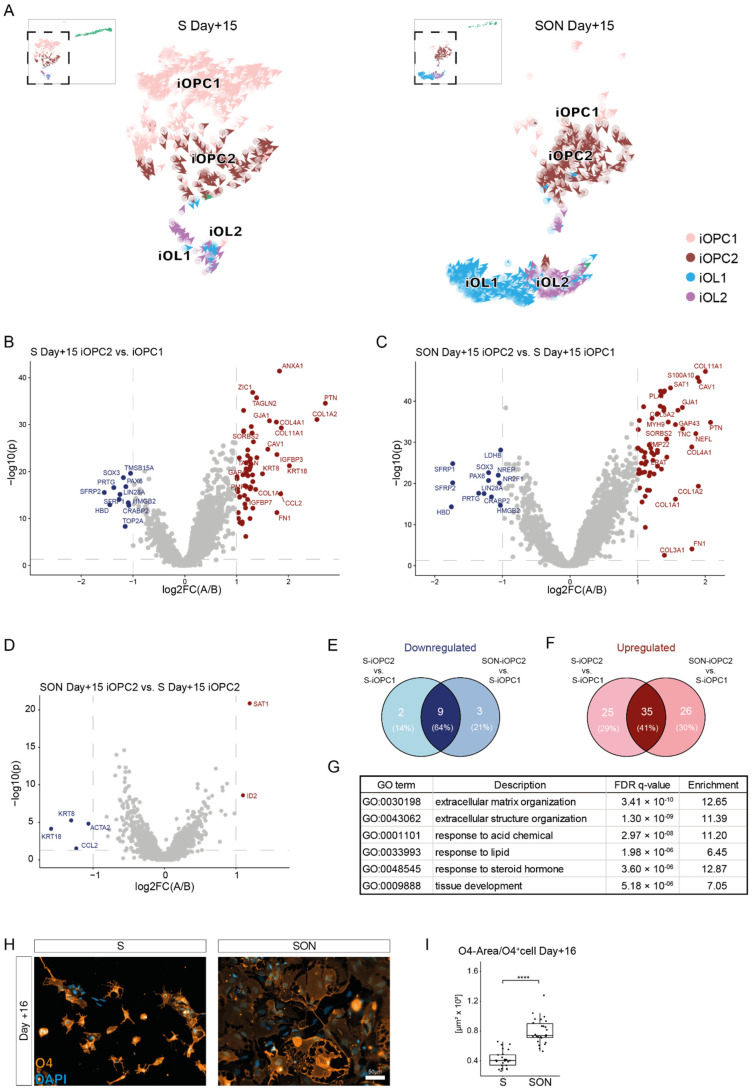
Characterization of iOPC subcluster formation induced by S- and SON-directed differentiation reveals profound differences in expression of extracellular matrix components. (**A**) Zoom-out of oligodendroglial lineage clusters of RNA velocity vectors projected on UMAP dimension plot of iOPC1, iOPC2, iOL1, and iOL2 clusters of S Day + 15 and SON Day + 15. For a complete dimension plot including neuronal linage, see Appendix A. (**B**) Differential gene expression (DGE) analysis of S-iOPC2 subcluster and S-iOPC1 subcluster illustrated with volcano plot. The average log2 fold change (log2FC) is shown on the *x*-axis and the negative log10 *p*-value (−log10(*p*)) on the *y*-axis. Genes significantly upregulated (red) and downregulated (blue) in S-iOPC2 compared with S-iOPC1 are colored. The top 20 differentially expressed genes are labeled for up- and downregulation accordingly. (**C**) DGE analysis of SON-iOPC2 subcluster and S-iOPC1 subcluster illustrated with a volcano plot. Genes significantly upregulated (red) and downregulated (blue) in SON-iOPC2 compared with S-iOPC1 are colored. (**D**) DGE analysis of SON-iOPC2 subcluster and S-iOPC2 subcluster illustrated with a volcano plot. Genes significantly upregulated (red) and downregulated (blue) in SON-iOPC2 are colored. (**E**) Upregulated genes within iOPC subclusters shown in a Venn diagram. (**F**) Downregulated genes within iOPC subclusters shown in a Venn diagram. (**G**) Pathway enrichment analysis of iOPC subcluster comparison. (**H**) Representative images of O4^+^ iOPCs after microbead purification. (**I**) Quantification of cell sizes of O4^+^ iOPCs illustrated with a box-and-whisker plot. Dots correspond to analyzed fields of view (*n* = 25–26 per condition). Statistical analysis by two-sided *t*-test. **** *p* < 0.0001.

## Data Availability

scRNAseq data (GEO: GSE179516) and codes for transcriptomic analyses (GitHub: https://github.com/MariusStephan/scRNAseq_S-SON, last accessed 1 December 2021) are publicly available.

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
