# Peer review of "Expression of Lineage Transcription Factors Identifies Differences in Transition States of Induced Human Oligodendrocyte Differentiation"

_cells, 2022, doi:10.3390/cells11020241_

Round 1

Reviewer 1 Report

I have no more questions.

Author Response

We like to thank the reviewer for his constructive feedback.

Reviewer 2 Report

The authors are to be congratulated on a true tour-de-force in quantitative analysis of the molecular-level differentiation cascade induced by different combinations of transcription factors used to promote oligodendrocyte differentiation from neural-induced iPSC cultures.  This degree of quantitative rigor has been missing from much of the literature.  The analyses are thorough and generally well presented.  Although the authors do mention it somewhat in the discussion, I would recommend greater emphasis on what it is that is actually being modeled here.  There are two principal reasons for making hiPSC-derived differentiated cells: modeling normal development or disease processes; and therapeutic use. As noted by the authors, some of these TF combinations can induce direct OL differentiation from fibroblasts; this is clearly an artificial and not a "physiological" process. Similarly, they point out that small molecule-induced differentiation can take 75-200 days, and that the advantage of using these S, SO or SON TF combinations is in speeding the differentiation process.  They do compare their in vitro-generated iOL clusters to normal murine OLs.  iOLs that have sufficiently authentic phenotype and function may be useful as therapeutic products, a very significant benefit; but it is not clear to what extent all of this painstakingly documented information would be relevant to natural, in vivo OL development and therefore to disease models.  I would simply suggest adding a bit more discussion on these points.  More minor issues: (1) the authors use their lentiviral vector at the hiPSC stage rather than after neural induction; the justification and advantage of this approach is not clear (2) there are some portions of the text that appear garbled, especially including Lines 79-85 of the introduction, and require proof reading.

Author Response

Overall Feedback:

Author response: We like to thank the reviewer first for his encouraging feedback on the overall quality of our study.

Reviewer 2 Comment: … iOLs that have sufficiently authentic phenotype and function may be useful as therapeutic products, a very significant benefit; but it is not clear to what extent all of this painstakingly documented information would be relevant to natural, in vivo OL development and therefore to disease models.  I would simply suggest adding a bit more discussion on these points.  

Author response: We have added the following paragraph to the very end of the discussion in which we aim to provide an outlook into possible applications in the future where indeed the ‘painstaking’ documentation of dynamic aspects of normal versus diseased profiling of oligodendrocytes may be helpful. We know that the aspects of the relevance of our study/applied tools towards a better understanding of the normal and diseased OL development could be discussed in more detail. Nonetheless, we think that the balance between the discussion of technical details and relevance for future applications is now a bit more balanced.

We believe that the molecular and analysis tools applied in this study to force and dissect human oligodendrocyte development may support the understanding of dynamic aspects of cell-autonomous versus non-cell-autonomous contributions in in vivo disease models in the future, i.e via grafting of iOLs generated from normal and diseased donors into mouse brains. The paradigmatic study on human iPSC-derived glia/mouse chimeras already revealed oligodendroglia-cell-autonomous contributions to childhood-onset schizophrenia [48]. More such in vivo studies are needed that combine patient-derived oligodendoglial cell types with sophisticated single-cell level analyses to further increase our knowledge of disorders with myelination deficits.

More minor issues: (1) the authors use their lentiviral vector at the hiPSC stage rather than after neural induction; the justification and advantage of this approach is not clear …

Author response: To make this point clearer, we have modified the following paragraph of the discussion in the following way:

‘To improve purity and scalability, we introduced the lentivirus-encoded lineage transcription factors at the level of iPSC cultivation followed by two puromycin selection steps, one before and one after initiation of iOPC/iOL differentiation; this protocol was robust for different iPSC lines obtained from six different human donors. The advantage of this approach is that less lentivirus is needed and only infected S/O/N-hiPSCs are proliferative and can be amplified, frozen, banked and recovered en masse at the time needed. In the future, this approach will be particularly suited to enable larger-scaled case-control comparisons and may allow also genetic and compound screenings requiring high numbers of dedicated cells.’

… (2) there are some portions of the text that appear garbled, especially including Lines 79-85 of the introduction, and require proof reading.

Author response: We thank the reviewer for pointing towards the last part of the introduction, which was indeed added after proof-reading by a professional board-approved editor (see Acknowledgments). The following part was adapted accordingly and now reads:

‘Therefore, we systematically compared S-, SO- and SON-directed differentiation of individual oligodendroglial lineage cells by using a streamlined protocol in which all TF combinations were expressed from an identical lentivirus backbone and all cell culture conditions were standardized. We show that S-, SO- and SON-directed differentiation were all sufficient to generate high yields of O4+ iOPCs, however, SON provided significant more yield than SO and S. Further investigations with S and SON show that SON allows an earlier generation of MBP+ iOLs with higher yields and more complex morphology. Subsequent scRNAseq experiments including RNA velocity analysis reveals a fastened directed oligodendroglial differentiation using SON and higher maturation stages of SON-iOLs compared to S-iOLs. We show that scRNAseq including RNA analysis are not limited to dissect a static stage but allows to dissect the time-dependent dynamics of directed differentiation and highlights that SON-directed differentiation might be better suited for research with a focus on more mature iOL.’

This manuscript is a resubmission of an earlier submission. The following is a list of the peer review reports and author responses from that submission.

Round 1

Reviewer 1 Report

The authors performed quantitative image analysis and single-cell transcriptomics to compare 3 transcription factor combinations for their efficacy towards oligodendrocyte lineage conversion. The results showed that Sox 10 alone generates a population of oligodendrocyte-precursor cells (OPCs) that were more immature than those generated by combination of 3 transcription factors and to display distinct molecular properties. 

Major comment: This is solid experimental work. Additional experiments are not required, but, in my opinion, the authors did not fully highlight what is the novelty of their work, what controversial issues they solved in their work. In the new version of the manuscript, It would be beneficial for the article if  in the abstract, introduction and discussion the authors discuss biological significance of results obtained.   The  conclusion "Our work highlights that TFs for generating iOPCs or iOLs should be chosen depending on the intended application or research question." seems too descriptive.

Minor comments

  1. In Materials and methods I did not find the information about the sourse and passage numbers of iPSCs.
  2. Figs 4 and S4 are "blind", nothing is visible in the pictures and the text is too small

Author Response

Response to reviewer 1:

Author Comment: We would like to thank the reviewer for feedback to improve the manuscript. Please note that all changes have been made with track changes on and are underlined and coloured in the revised manuscript to ease identification of changed parts.

Rev 1 comment: The authors performed quantitative image analysis and single-cell transcriptomics to compare 3 transcription factor combinations for their efficacy towards oligodendrocyte lineage conversion. The results showed that Sox 10 alone generates a population of oligodendrocyte-precursor cells (OPCs) that were more immature than those generated by combination of 3 transcription factors and to display distinct molecular properties. 

Major comment: This is solid experimental work. Additional experiments are not required, but, in my opinion, the authors did not fully highlight what is the novelty of their work, what controversial issues they solved in their work. In the new version of the manuscript, It would be beneficial for the article if in the abstract, introduction and discussion the authors discuss biological significance of results obtained.   The conclusion "Our work highlights that TFs for generating iOPCs or iOLs should be chosen depending on the intended application or research question." seems too descriptive.

Author Reply: We have tried to improve the manuscript accordingly. Nonetheless, the presented study is a technical report, and we think that these aspects should be put forward as the main topic.

Rev 1 comment: Minor comments

  1. In Materials and methods I did not find the information about the sourse and passage numbers of iPSCs.

Author Reply: We thank the reviewer for having identified this lack of essential information. We have modified chapter 2.2. (now “hiPSC lines, hiPSC cultivation and lentiviral transfection”) and added a paragraph concerning the used hiPSC lines and their characterization. Moreover, we provided information (age, sex, passage number) regarding the used hiPSC lines in Table S1 in the supplemental information.

  1. Figs 4 and S4 are "blind", nothing is visible in the pictures and the text is too small

Author Reply: We do agree and have modified the figures accordingly.

Reviewer 2 Report

In this manuscript, Raabe et al. constructed a lentiviral-based inducible overexpression system to study the role of key transcriptional factors in human oligodendrocyte differentiation. They specifically evaluated SOX10, OLIG2, NKX6-2 and different combinations after neural progenitor induction. They found SOX10 only induced oligodendrocyte-precursor cells, while SOX10/OLIG2/NKX6-1 together could efficiently generate mature oligodendrocyte. Single-cell RNA-seq data also supported the conclusion. This study is generally solid and informative for understanding the role of the individual specific transcription factor in oligodendrocyte lineage differentiation, although the conclusion is not totally unexpected.

Another major concern is the inducible overexpression system. From Figure 1D, the FLAG-positive cells are below a half for the SON vector. A common question is, what is the percentage of generated oligodendrocytes is exactly due to the successful overexpression of ectopic transcription factors (i.e., FLAG-positive)? How to understand the differentiated oligodendrocytes with no ectopic overexpression, and the non-oligodendrocytes but with successful SON expression? The authors have the single cell data and are able to analyze the heterogeneity.

In addition, I have one question for the UMAP of single cell analysis. The NPC population is extremely separated from other population, and there seems no link with iOPC. How to understand this point, given that all the iOPC and iOL are derived from NPC like the neuronal population?

Author Response

Response to reviewer 2:

Author Comment: We would like to thank the reviewer for the feedback to improve the manuscript. Please note that all changes have been made with track changes on and are underlined and coloured in the revised manuscript to ease identification of changed parts.

Rev 2 comment: In this manuscript, Raabe et al. constructed a lentiviral-based inducible overexpression system to study the role of key transcriptional factors in human oligodendrocyte differentiation. They specifically evaluated SOX10, OLIG2, NKX6-2 and different combinations after neural progenitor induction. They found SOX10 only induced oligodendrocyte-precursor cells, while SOX10/OLIG2/NKX6-1 together could efficiently generate mature oligodendrocyte. Single-cell RNA-seq data also supported the conclusion. This study is generally solid and informative for understanding the role of the individual specific transcription factor in oligodendrocyte lineage differentiation, although the conclusion is not totally unexpected. 

Another major concern is the inducible overexpression system. From Figure 1D, the FLAG-positive cells are below a half for the SON vector. A common question is, what is the percentage of generated oligodendrocytes is exactly due to the successful overexpression of ectopic transcription factors (i.e., FLAG-positive)? How to understand the differentiated oligodendrocytes with no ectopic overexpression, and the non-oligodendrocytes but with successful SON expression? The authors have the single cell data and are able to analyze the heterogeneity.

Author Reply: We do agree with the reviewer that the lentiviral approach is limited by generating most likely an increased level of heterogeneity of trans-differentiated cells that may originate from different construct expression levels. Please note that we have discussed this in the second paragraph of the discussion already and briefly interpreted the construct expression in the results part (3.8). Moreover, we extended the analysis and discussion regarding construct expression levels. Please be aware, that the strength of scRNAseq data is the single cell resolution and that scRNAseq data are generally noisier and less deep (we included all cells with at least 1000 detected features) compared to bulk RNAseq. We think that sensitivity issues both for the detection of the FLAG epitope (particularly of SON constructs) (Figure 1C,D) and for expression levels of exogenous transcripts in single cell transcriptomes (Figure S6D) would strongly bias any attempts for further downstream analysis. (i) The dramatic drop of cleavage efficiencies by single versus double P2A/T2A element containing constructs (e.g. Liu Z et al., Systematic comparison of 2A peptides for cloning multi-genes in a polycistronic vector. Sci Rep. 2017 May 19;7(1):2193.) (together with the larger size of the SON vs S construct) may simply explain the drop in FLAG+ cells. We addressed this now in the discussion. (ii) In addition, we like to draw the attention of the Reviewer to the well-known problem of detecting exogenous mRNAs expressed from lentiviral constructs i.e. in Perturb-Seq approaches that can be partially overcome by applying a second ‘construct specific step-out’ PCR to sufficiently enrich these transcripts to finally improve single cell annotations (see Dixit A et al., Perturb-Seq: Dissecting Molecular Circuits with Scalable Single-Cell RNA Profiling of Pooled Genetic Screens. Cell. 2016 Dec15;167(7):1853-1866.e17., and many follow up applications). Since we do not have this additional data, we believe that any attempt to try to extract meaningful biological insight by separating i.e. high versus low S or SON expressing cells from the data sets we have generated, will be of very limited value. Nonetheless, we have modified Figure S6 and added S6D, where we display the level of the exogenous S and SON expression in cells of the respective subclusters to at least better illustrate the heterogeneity obtained with the Lentiviral approach.

Rev 2 comment: In addition, I have one question for the UMAP of single cell analysis. The NPC population is extremely separated from other population, and there seems no link with iOPC. How to understand this point, given that all the iOPC and iOL are derived from NPC like the neuronal population?

Author Reply: Our explanation for the strict separation of the NPC cluster (that derived from sample NI Day+0 without doxycycline) and the two differentiated samples of S/SON Day+15, is the extended time window (15 days) of the sample collection. Therefore, we do not capture all intermediate stages (as the intermediate/immature cells connecting NPCs and Neurons in the NPC sample).

Round 2

Reviewer 2 Report

I do agree this study provides an informative resource, however, I don't think the authors have fully addressed my questions, particularly about the significance (of the major finding and the scRNA-seq data). The technical concern also greatly limits the value of mechanistic intepretation based on scRNA-seq.